# MASKED AUTOENCODERS ARE ROBUST NEURAL ARCHITECTURE SEARCH LEARNERS?

## ABSTRACT

Neural Architecture Search (NAS) currently relies heavily on labeled data, which is both expensive and time-consuming to acquire. In this paper, we propose a novel NAS framework based on Masked Autoencoders (MAE) that eliminates the need for labeled data during the search process. By replacing the supervised learning objective with an image reconstruction task, our approach enables the robust discovery of network architectures without compromising performance and generalization ability. Additionally, we address the problem of performance collapse encountered in the widely-used Differentiable Architecture Search (DARTS) method in the unsupervised paradigm by introducing a multi-scale decoder. Through extensive experiments conducted on various search spaces and datasets, we demonstrate the effectiveness and robustness of the proposed method, providing empirical evidence of its superiority over baseline approaches.

## 1 INTRODUCTION

In recent years, there has been a significant surge of interest in Neural Architecture Search (NAS) within the field of machine learning (Zela et al., 2020; Liang et al., 2019). NAS algorithms have emerged as a powerful tool for automatically discovering superior network architectures, potentially saving valuable time and effort for human experts. These algorithms have demonstrated remarkable success in various tasks, including but not limited to image classification and object detection, by discovering architectures that achieve state-of-the-art results.

Existing NAS methods focus on learning from labeled data, leveraging the power of supervised learning to guide the search for optimal network architectures. By utilizing labeled data, which consists of input samples paired with their corresponding ground truth labels, NAS algorithms seek to discover architectures that can accurately classify or predict various types of data. However, obtaining substantial quantities of human-annotated data proves to be costly and time-consuming. A portion of the research (Liu et al., 2020; Yan et al., 2020; Zhang et al., 2021) has shifted its attention towards exploring methods to minimize the reliance on annotated data.

In this study, we present a novel NAS framework based on MAE (He et al., 2022) (named MAE-NAS), an area that has received limited explicit exploration in prior research to the best of our knowledge. We apply the searching strategy with the widely adopted DARTS method. Instead of relying on the supervised learning objective employed in DARTS, we replace it with the image reconstruction loss, thereby obviating the need for labeled data during the search process. This approach draws inspiration from SimMIM (Xie et al., 2022), which has demonstrated remarkable performance in image classification by learning visual representations. Specifically, we randomly generate masked images and search for the model that can accurately reconstruct the original input image. In this way, DARTS can effectively and robustly discover promising network architectures without being reliant on labeled data. In contrast to supervised NAS methods, which often suffer from overfitting and lack generalization ability despite achieving near-zero training error, MAE-NAS can discover models with enhanced representation and improved generalization ability.

Based on the MAE-NAS, we conduct further investigation into the issue of performance collapse in DARTS within an unsupervised setting. We observe an intriguing phenomenon wherein the occurrence of collapse is highly correlated with the size of the mask ratio. Notably, a higher mask ratio (i.e., greater than 0.5) effectively enables DARTS to robustly overcome performance collapse. This observation is indeed reasonable: given DARTS' susceptibility to unstable training and the tendency

to become trapped in local minima, the mask ratio can be viewed as a form of regularization. To address this issue at its core, we propose the utilization of a multi-scale decoder to stabilize the training process and prevent collapse. Specifically, the decoder takes multi-scale features of DARTS as input, which encode both fine and coarse-grained information of the image. These features are subsequently upsampled and combined using a linear layer to generate the final reconstructed image.

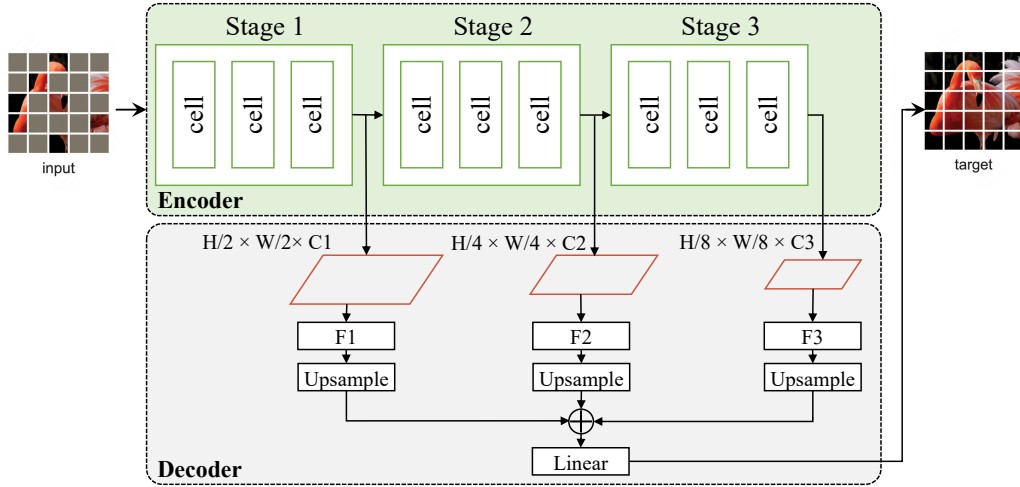

Figure 1: The searching pipeline of MAE-NAS. $F_1$ to $F_3$ are channel alignment modules implemented with a simple convolution.

The effectiveness of our method has been verified on seven widely used search spaces and three datasets, providing compelling empirical evidence. Experimental results on ImageNet (Deng et al., 2009) demonstrate that MAE-NAS achieves comparable Top-1 accuracy while adhering to the same complexity constraint and search space. Furthermore, we have conducted comprehensive experimental analysis and ablation studies to gain a deeper understanding of the characteristics of our proposed method using the NASBench-201 (Deng et al., 2009). These analyses reveal that masked autoencoders are robust neural architecture search learners compared to their baseline counterparts. In summary, the main contributions of this work can be summarized as follows:

- We present a novel NAS framework that leverages Masked Autoencoder to enable label-free searching, which addresses the challenge of NAS in scenarios where labeled data is not readily available.

- The proposed method is designed to be plug-and-play, seamlessly integrating with existing supervised NAS methods. In our experiments, we showcase the compatibility of our approach with other orthogonal DARTS variants. By removing their handcrafted indicators, our method demonstrates its ability to integrate without incurring any additional overhead.

- Our approach achieves comparable results to its supervised counterpart. Additionally, our approach excels in scenarios where the supervised method encounters difficulties, such as resolving the problem of performance collapse in DARTS.

## 2 RELATED WORK

Manually designed neural networks have demonstrated remarkable success across various computer vision tasks. However, it is widely acknowledged that these artificial architectures may not be optimal. Consequently, there has been a growing interest in NAS from both academic and industrial communities.

**Supervised neural architecture search**. Supervised neural architecture search has emerged as a prominent paradigm in NAS research. Initially, NAS methods involved training candidate architectures from scratch and iteratively updating the controller based on performance feedback. However,

this approach incurred a substantial computational cost, as exemplified by NAS-Net (Zoph et al., 2018b), which required approximately 1350-1800 GPU days. To address this challenge and enhance the efficiency of NAS, weight-sharing mechanisms have been widely adopted in various studies. These approaches can also be classified into two main categories: one-shot methods (Bender et al., 2018; Dong & Yang, 2019a) and gradient-based methods (Liu et al., 2019; Chu et al., 2021).

One-shot methods (Bender et al., 2018) entail training an over-parameterized supernet using diverse sampling strategies. Once the supernet is effectively trained, multiple child models are evaluated as potential alternatives, and those exhibiting superior performance are selected. In contrast, gradient-based algorithms optimize both the network weights and architecture parameters simultaneously through back-propagation. The selection of operators is based on the magnitudes of the architecture parameters. These approaches aim to reduce the computational cost of NAS while still achieving commendable performance. Through the utilization of weight-sharing mechanisms and the adoption of different optimization strategies, researchers have made significant progress in enhancing the efficiency and practicality of NAS. Leveraging the differentiable and end-to-end characteristics of DARTS, we adopt the DARTS paradigm to investigate unsupervised NAS in our study.

**Unsupervised neural architecture search**. In recent years, there has been a growing emphasis on the application of unsupervised learning in various domains, including the field of NAS. This unsupervised paradigm has gained attention due to its potential to alleviate the reliance on labeled data. Notably, the work of in UnNAS (Liu et al., 2020) provides a comprehensive analysis of the impact of labeled data on NAS performance. Their findings challenge the conventional belief that labeled data is indispensable for NAS. Building upon this, RLNAS (Zhang et al., 2021) leverages random labels instead of true labels. Surprisingly, their research demonstrates that neural architectures discovered using random labels can achieve comparable or even superior performance to supervised NAS methods. In contrast to these existing approaches, our study takes a distinct approach to unsupervised NAS.

## 3 METHOD

As stated in the introduction, to leverage the DARTS mechanism (Liu et al., 2019), we will first provide a concise overview of DARTS as a foundational step. Building upon the DARTS framework, we will introduce our proposed approach MAE-NAS.

### 3.1 PRELIMINARY: DARTS

DARTS (Liu et al., 2019) has emerged as a groundbreaking technique in the area of neural architecture search, signifying a remarkable advancement. The search space within DARTS is composed of multiple cells organized in stacks, where each cell is depicted as a directed acyclic graph. These cells encompass a sequence of nodes, each representing a latent feature map. The connections between nodes assume a pivotal role, as they represent operations selected from a diverse range of candidates, including convolution, pooling, zero, skip-connect, and more. We denote the operation from node $i$ to $j$ as $o^{i,j}$. The output of each intermediate node is computed by considering all of its predecessors, as expressed by the equation $x_j = \sum_{i<j} o^{i,j}(x_i)$. DARTS introduces an innovative approach by transforming the pursuit of the optimal architecture into a quest for the best operation on each connection. To ensure a continuous search space, it relaxes the categorical decision of a specific operation and instead considers a weighted sum encompassing all possible operations. We denote the architecture parameter between node $i$ and node $j$ as $\alpha_k^{i,j}$, with the corresponding operation being $o_k$. DARTS employs the softmax function to assign weights to various operations. This enables the model to determine the importance of different operations within the architecture. The output of each intermediate node is calculated by considering the contributions from all of its preceding nodes.

Let $\mathcal{L}_{train}$ and $\mathcal{L}_{val}$ represent the training and validation loss, respectively. Both losses are influenced by the architecture parameter $\alpha$ and the weights $w$ in the network. The goal of architecture search is to find $\alpha^*$ that minimizes the validation loss $\mathcal{L}_{val}(w^*, \alpha^*)$, where the weights $w^*$ associated with the architecture are obtained by minimizing the training loss $w^* = \arg\min_w \mathcal{L}_{train}(w, \alpha^*)$.

This formulation leads to a two-level optimization problem:

$$
\begin{aligned}
\min_{\alpha} \quad & \mathcal{L}_{val}(w^*(\alpha), \alpha) \\
s.t. \quad & w^*(\alpha) = \arg\min_{w} \mathcal{L}_{train}(w, \alpha)
\end{aligned}
\tag{1}
$$

## 3.2 OUR APPROACH: DARTS BASED ON MASKED AUTOENCODERS

Our approach is grounded in a crucial observation: supervised neural architecture search often yields final models that overfit the training data. In other words, regardless of how we optimize $\alpha$ and $w$ in Eq.1, these models consistently achieve near-zero training error. However, the ultimate goal of the search process is to identify architectures that exhibit strong generalization performance on the validation set. This presents an inherent contradiction in supervised learning. With this perspective in mind, we propose leveraging the widely-used SimMIM (Xie et al., 2022) as a proxy task for NAS. The aim is to discover models with enhanced generalization capabilities. Building upon the DARTS mechanism, our new optimization objective can be formulated as follows:

$$
\begin{aligned}
\min_{\alpha} \quad & \mathcal{L}_{val}(w^*(\alpha), \alpha, M) \\
s.t. \quad & w^*(\alpha) = \arg\min_{w} \mathcal{L}_{train}(w, \alpha^*, M)
\end{aligned}
\tag{2}
$$

where $M$ represents the set of masked pixels. Following the SimMIM framework, our method comprises an encoder that transforms the observed signal into a latent representation, as well as a decoder that reconstructs the original signal from this latent representation. Specifically, DARTS serves as the backbone of the encoder. In this way, the framework of the masked autoencoder becomes robust NAS learners, seeking to learn promising encoder structures from the DARTS space, resulting in minimal image reconstruction errors.

### 3.2.1 ESCAPE FROM PERFORMANCE COLLAPSE

DARTS exhibits a significant decline in performance when skip connections become dominant in a supervised setting. Numerous studies (Chu et al., 2021; Xu et al., 2020) have shed light on the underlying cause of this behavior. It is attributed to the unstable training of the supernet, resulting in unfair competition between skip connections and other operations. Consequently, several approaches (Chu et al., 2021; Zela et al., 2020) have been proposed to address this issue by introducing various types of regularization to facilitate DARTS in escaping local optima and achieving better generalization properties. However, it remains unclear whether this problem also exists in unsupervised NAS.

To investigate this matter, we conduct three iterations of MAE-NAS using different random seeds. Subsequently, we identified the top eight performing operations for each cell, with two operations assigned to each of the four intermediate nodes. In this context, a dominant operation refers to the one with the highest softmax value among all the candidates for incoming edges of a specific node. The results are presented in Table 1. Interestingly, we observe a phenomenon in unsupervised NAS where the occurrence of collapse is highly correlated with the size of the mask ratio. Specifically, when the mask ratio is less than 0.5, the probability of collapse is significantly high, whereas when the mask ratio exceeds 0.5, the occurrence of collapse is almost negligible.

Remarkably, this finding aligns with the conclusions drawn from supervised learning. The unstable training process of DARTS makes it prone to converging towards sharper optima, resulting in performance collapse. R-DARTS (Zela et al., 2020) addresses this problem by introducing various regularizations, such as $L_2$ or ScheduledDropPath regularization, enabling it to escape local optima. In a sense, the mask ratio can be regarded as a form of regularization that robustly helps DARTS overcome performance collapse.

### 3.2.2 HIERARCHICAL DECODER DESIGN

Adjusting the mask ratio seems to be a solution. However, it is important to note that improper threshold settings can lead to the unjust rejection of promising architectures. In order to fundamentally address

Table 1: The average number of dominant skip connections with different mask ratio in three independent experiments.

| Mask Ratio | 0.2 | 0.4 | 0.6 | 0.8 |
|---|---|---|---|---|
| No. of skips | 6 | 5 | 2 | 1 |

this issue, we propose the utilization of a multi-scale decoder, which serves to stabilize the training process and prevent the occurrence of the collapse phenomenon.

In the SimMIM framework, the decoder takes the tokens derived from the encoder as inputs, and subsequently processes them through a series of transformer blocks to reconstruct the image. In contrast, our encoder (i.e. DARTS) is designed to extract multi-scale features, denoted as $F_1$, $F_2$, and $F_3$ which encode both fine-grained and coarse-grained information pertaining to the image. To supervise the training of these multi-grained representations, we upsample $F_1$, $F_2$ and $F_3$ to match the size of the input image, respectively. Subsequently, we combine these multi-grained features using a linear layer, resulting in the final reconstructed image. This process can be mathematically represented as follows:

$$F_d = Linear(Upsample(Conv(F_1), 2) + Upsample(Conv(F_2), 4) + Upsample(Conv(F_2), 8)) \tag{3}$$

The multi-scale decoder, depicted in the bottom-left section of Figure 1, is responsible for facilitating searching process. In terms of the objective function, we adopt the same loss functions as SimMIM, which are employed to reconstruct the masked image patches. Furthermore, we solely focus on the reconstruction of these masked patches within the objective function, disregarding other regions of the image.

### 3.2.3 UNDERSTANDING FROM DIFFUSION MODEL VIEW

Diffusion models (Ho et al., 2020; Dhariwal & Nichol, 2021) belong to the category of probabilistic generative models that aim to generate high-quality images from random noise through iterative steps. By leveraging the inherent probabilistic nature of these models, they are capable of capturing complex patterns and structures present in the image data. Through each iteration, the diffusion model gradually refines the generated image by reducing noise and enhancing details. This iterative process enables the generation of visually appealing and realistic images that closely resemble the original data. In fact, when the mask ratio is set to 100%, image restoration based on SimMIM can also be viewed as a diffusion process. Specifically, Gaussian noise is added to the masked patches, and the generation process gradually removes the random noise until the masked patches are reconstructed.

Within the diffusion model, the encoder plays a critical role in reconstructing high-quality images. To enhance the generation quality, an end-to-end system is designed to learn a more powerful encoder backbone. The encoder is responsible for capturing and encoding essential features and information from the input data. By improving the capabilities of the encoder, the system can extract more meaningful representations and generate more accurate and visually appealing images. This upgrade in the encoder backbone significantly contributes to the overall performance and effectiveness of the diffusion model in image generation tasks.

## 4 EXPERIMENTS

### 4.1 SEARCH SPACES AND TRAINING DETAILS

Comprehensive experiments are conducted on several popular architectural design spaces, such as NASBench-201 (Dong & Yang, 2020), DARTS-based. Following the experiment settings in DARTS- (Chu et al., 2021), we apply the searching, training, and evaluation procedure on the standard DARTS search space (named $S0$). For other DARTS-like search spaces ($S1$-$S4$) proposed in R-DARTS (Zela et al., 2020), we follow the same setting as the original paper. As the comparison method, S-DARTS (Chen & Hsieh, 2020) differs from R-DARTS in layers and initial channels for training from scratch on CIFAR-100. For a fair comparison, we align such two training settings respectively. Besides, the effectiveness of our approach is evaluated on NASBench-201, which is built for benchmarking NAS algorithms. For ImageNet, our method apply PC-DARTS (Xu et al., 2020) to search on the standard DARTS search space. The retraining setting follows MobileNetV3 (Howard et al., 2019). For mask image modeling, the mask ratio is simply set to 0.5. The patch size of the mask is 8 and 4 respectively for ImageNet and CIFAR.

Table 2: CIFAR-10 results on DARTS search space. The average results of 5 independently experiments are reported.

| Models | Params (M) | FLOPs (M) | Top-1 Acc (%) | Cost (GPU Days) |
|---|---|---|---|---|
| NASNet-A (Zoph et al., 2018a) | 3.3 | 608 | 97.35 | 2000 |
| ENAS (Pham et al., 2018) | 4.6 | 626 | 97.11 | 0.5 |
| DARTS (Liu et al., 2019) | 3.3 | 528 | 97.00±0.14 | 0.4 |
| SNAS (Xie et al., 2019) | 2.8 | 422 | 97.15 0.02 | 1.5 |
| GDAS (Dong & Yang, 2019b) | 3.4 | 519 | 97.07 | 0.2 |
| P-DARTS (Chen et al., 2019) | 3.4 | 532 | 97.5 | 0.3 |
| PC-DARTS (Xu et al., 2020) | 3.6 | 558 | 97.43 | 0.1 |
| DARTS- (best) (Chu et al., 2021) | 3.5 | 568 | **97.5** | 0.4 |
| **Ours (best)** | 3.8 | 605 | 97.48 | 0.4 |
| P-DARTS(Chen et al., 2019) | 3.3± 0.21 | 540±34 | 97.19±0.14 | 0.3 |
| R-DARTS (Zela et al., 2020) | - | - | 97.05±0.21 | 1.6 |
| DARTS- (avg) (Chu et al., 2021) | 3.5±0.13 | 583±22 | 97.41±0.08 | 0.4 |
| **Ours (avg)** | 4.05±0.23 | 639 ±34 | **97.43±0.05** | 0.4 |

Table 3: Search results on ImageNet. The top block indicates the architectures are searched on CIFAR-10 and trained from scratch on ImageNet. Other blocks search and train models on ImageNet. The bottom block shows some unsupervised NAS methods.

| Models | FLOPs (M) | Params (M) | Top-1 Acc (%) | Top-5 Acc (%) | Cost (GPU Days) |
|---|---|---|---|---|---|
| NASNet-A (Zoph et al., 2018a) | 564 | 5.3 | 74.0 | 91.6 | 2000 |
| DARTS (Liu et al., 2019) | 574 | 4.7 | 73.3 | 91.3 | 0.4 |
| SNAS (Xie et al., 2019) | 522 | 4.3 | 72.7 | 90.8 | 1.5 |
| PC-DARTS (Xu et al., 2020) | 586 | 5.3 | 74.9 | 92.2 | 0.1 |
| FairDARTS-B (Chu et al., 2020) | 541 | 4.8 | 75.1 | 92.5 | 0.4 |
| AmoebaNet-A (Real et al., 2019) | 555 | 5.1 | 74.5 | 92.0 | 3150 |
| MnasNet-92 (Tan et al., 2019) | 388 | 3.9 | 74.79 | 92.1 | 3791 |
| FBNet-C (Wu et al., 2019) | 375 | 5.5 | 74.9 | 92.3 | 9 |
| FairNAS-A (Chu et al., 2019) | 388 | 4.6 | 75.3 | 92.4 | 12 |
| PC-DARTS (Xu et al., 2020) | 597 | 5.3 | 75.8 | 92.7 | 3.8 |
| DARTS- (Chu et al., 2021) | 467 | 4.9 | **76.2** | 93.0 | 4.5 |
| **Ours** | 533 | 4.7 | 76.11 | 92.8 | 4.5 |

## 4.2 SEARCHING ON CIFAR-10

As shown in Table 2, regardless of whether it is the optimal or average result, the architectures found by our method perform well on CIFAR-10 (Krizhevsky et al., 2009). It is worth emphasizing that our method does not require labels while achieving comparable even better performance with other supervised methods. Besides, the search cost is 0.4 GPU day, which is not higher than other methods. Such improvement is probably due to the fact that the architectures found by our method have more flops. But it's reasonable that models with higher flops are more likely to have the better capability if the flops are not constrained.

## 4.3 SEARCHING ON IMAGENET

**Comparison with supervised NAS methods**. To thoroughly verify the effectiveness of MAE-NAS, we perform searching directly on ImageNet (Deng et al., 2009) in S0. From Table 3, our approach achieves 76.11% top-1 accuracy, which outperforms the searched models on CIFAR-10 with a clear margin. Besides, MAE-NAS is not inferior to searched models on ImageNet by supervised approaches. The above results fully demonstrate the potential of masked autoencoders as a proxy task in the NAS area.

Table 4: Comparison with unsupervised NAS methods on ImageNet.

| Method | FLOPs (M) | Params (M) | Top-1 Acc (%) | Top-5 Acc (%) |
|---|---|---|---|---|
| UnNAS (rotation task) | 552 | 5.1 | 75.8 | 92.6 |
| UnNAS (color task) | 587 | 5.3 | 75.5 | 92.6 |
| UnNAS ((jigsaw task) | 560 | 5.2 | 76.2 | 92.8 |
| RLNAS | 561 | 5.2 | 75.9 | 92.8 |
| **Ours** | 533 | 4.7 | 76.11 | 92.8 |

Table 5: Search results on NAS-Bench-201. To give an objective evaluation of NAS algorithms, the best and average results are reported. The latter is computed by 4 runs of search.

| Method | Cost (hours) | CIFAR-10 valid | CIFAR-10 test | CIFAR-100 valid | CIFAR-100 test | ImageNet16-120 valid | ImageNet16-120 test |
|---|---|---|---|---|---|---|---|
| DARTS$^{1st}$ | 3.2 | 39.77±0.00 | 54.30±0.00 | 15.03±0.00 | 15.61±0.00 | 16.43±0.00 | 16.32±0.00 |
| DARTS$^{2st}$ | 10.2 | 39.77±0.00 | 54.30±0.00 | 15.03±0.00 | 15.61±0.00 | 16.43±0.00 | 16.32±0.00 |
| GDAS (2019b) | 8.7 | 89.89±0.08 | 93.61±0.09 | 71.34±0.04 | 70.70±0.30 | 41.59±1.33 | 41.71±0.98 |
| SETN (2019a) | 9.5 | 84.04±0.28 | 87.64±0.00 | 58.86±0.06 | 59.05±0.24 | 33.06±0.02 | 32.52±0.21 |
| DARTS- (avg) | 3.2 | 91.03±0.44 | 93.80±0.40 | 71.36±1.51 | 71.53±1.51 | 44.87±1.46 | 45.12±0.82 |
| DARTS- (best) | 3.2 | 91.55 | 94.36 | 73.49 | 73.51 | 46.37 | 46.34 |
| **Ours (avg)** | 3.2 | 90.67±0.57 | 93.77±0.53 | **71.4±2.0** | **71.75±1.75** | 43.15±3.15 | 43.77±2.53 |
| **Ours (best)** | 3.2 | **91.55** | **94.36** | **73.49** | **73.51** | **46.37** | **46.34** |
| optimal | n/a | 91.61 | 94.37 | 73.49 | 73.51 | 46.77 | 47.31 |

**Comparison with unsupervised NAS methods**. Table 4 futher gives comparison experiments with some unsupervised NAS methods, and MAE-NAS achieves comparable even better performace.

## 4.4 SEARCHING IN NAS-BENCH-201

NAS-Bench-201 (Dong & Yang, 2020) shares a similar skeleton as DARTS and differs from DARTS in the number of layers and nodes. Importantly, the search space trains 15625 models from scratch and provides their ground-truth performance, which allows researchers to focus on the search algorithms itself without unnecessary repetitive training of searched models. As shown in Table 5, search results on NASBench-201 further verify the superiority of MAE-NAS compared with supervised NAS methods. First, MAE-NAS helps the native DARTS get rid of the problem of collapse. Second, our approach sets a new state of the art on multiple datasets, approaching the optimal solution of the whole search space.

## 4.5 COMBINATION WITH OTHER VARIANTS

In this part, we verify the power of our approach combined with existing NAS algorithms. We choose two popular NAS algorithms (P-DARTS and PC-DARTS), whose public codes are available, to apply mask autoencoders as a proxy for further improvements. All experiments are conducted on ImageNet. The original training set is split into two parts: 50,000 images for validation and the rest for training.

**P-DARTS** The motivation behind P-DARTS (Chen et al., 2019) is to close the depth gap between searching and training neural architecture search by introducing a progressive search strategy. The method starts with a small network and progressively increases its size and complexity over multiple stages. Meanwhile, some prior knowledge is introduced for search space regularization, to get rid of the issue of collapse. For

Table 6: P-DARTS and its combination with ours on CIFAR-10. The manual tricks are removed in our experiments.

| Method | Setting | Top-1 Accuracy (%) |
|---|---|---|
| P-DARTS | w/o tricks | 96.48±0.55 |
| MAE-NAS | w/o tricks | 97.16±0.14 |

Table 7: PC-DARTS and its combination with ours on CIFAR-10.. Searching is repeated three times for average.

| Method | Top-1 Accuracy (%) | Cost |
|---|---|---|
| PC-DARTS | 97.09±0.14 | 3.75h |
| MAE-NAS | 97.27 | 3.41h |

example, they apply dropout after each skip-connect

operation. Besides, they control the number of preserved skip-connects manually. The aforementioned strategies, to some extent, compromise the fairness of the comparison. To this end, we remove these artificial limitations for fair comparison. We run P-DARTS without handcrafted tricks and our approach each three times to have an average result. As shown in Table 6, our approach achieves 97.16% Top-1 accuracy, which is 0.68% higher than P-DARTS. From this perspective, our method effectively mitigates the problem of collapse for P-DARTS without man-made prior.

**PC-DARTS** The motivation behind PC-DARTS (Partial Channel Connections for Memory-Efficient Differentiable Architecture Search) (Xu et al., 2020) is to address the challenges of memory and computational efficiency in neural architecture search. Traditional methods for architecture search require a large number of parameters and operations, making them computationally expensive and memory-intensive. PC-DARTS proposes a novel approach that reduces the number of parameters and operations required for architecture search, while maintaining high accuracy. The method uses partial channel connections, which allows for the sharing of parameters across different channels in a convolutional neural network. This reduces the number of parameters required for architecture search, while also reducing the computational cost.

To verify the effectiveness of masked autoencoders as a NAS proxy under the PC-DARTS setting, we compare the performance of PC-DARTS with its combination with ours. To ensure the reproducibility of our results, we utilized the code released by the authors and conducted multiple experiments with different random seeds under the same experimental settings. As Table 7, our method achieves a 0.18% increase in accuracy compared to PC-DARTS.

Overall, the above results demonstrate the potential of our method to enhance the performance of existing neural architecture search algorithms, even under suboptimal configurations. We believe that our approach can be further optimized and applied to a wide range of applications in the NAS field, paving the way for more efficient and effective neural architecture search in the future.

Table 8: Comparison on various datasets and search spaces. The lowest error rate of 3 found architectures is reported.

| Benchmark | | DARTS | R-DARTS | | DARTS | | DARTS- | Ours | PC-DARTS | SDARTS | | DARTS- | Ours[†] |
|---|---|---|---|---|---|---|---|---|---|---|---|---|---|
| | | | DP | L2 | ES | ADA | | | | RS | ADV | | |
| C10 | S1 | 3.84 | 3.11 | 2.78 | 3.01 | 3.10 | 2.68 | 2.92 | 3.11 | 2.78 | 2.73 | 2.68 | 2.92 |
| | S2 | 4.85 | 3.48 | 3.31 | 3.26 | 3.35 | 2.63 | 2.66 | 3.02 | 2.75 | 2.65 | 2.63 | 2.66 |
| | S3 | 3.34 | 2.93 | 2.51 | 2.74 | 2.59 | 2.42 | 2.50 | 2.51 | 2.53 | 2.49 | 2.42 | 2.50 |
| | S4 | 7.20 | 3.58 | 3.56 | 3.71 | 4.84 | 2.86 | **2.71** | 3.02 | 2.93 | 2.87 | 2.86 | **2.71** |
| C100 | S1 | 29.46 | 25.93 | 24.25 | 28.37 | 24.03 | 22.41 | 23.79 | 18.87 | 17.02 | 16.88 | 16.92 | 17.75 |
| | S2 | 26.05 | 22.30 | 22.24 | 23.25 | 23.52 | 21.61 | 22.58 | 18.23 | 17.56 | 17.24 | 16.14 | 17.13 |
| | S3 | 28.90 | 22.36 | 23.99 | 23.73 | 23.37 | 21.13 | 21.36 | 18.05 | 17.73 | 17.12 | 15.86 | 16.49 |
| | S4 | 22.85 | 22.18 | 21.94 | 21.26 | 23.20 | 21.55 | 21.84 | 17.16 | 17.17 | 15.46 | 17.48 | 16.54 |

## 4.6 ROBUSTNESS ON MULTIPLE SEARCH SPACES AND DATASETS

To validate the robustness of our proposed method, we conduct comparative experiments across four search spaces (S1-S4), two datasets (CIFAR-10, CIFAR-100), and multiple state-of-the-art (SOTA) methods (DARTS-, PC-DARTS etc.). As the search process of many NAS methods is not always stable, to ensure the fairness of our experiments, we independently repeated each experiment three times and took the average of the results. As shown in Table 8, without labels, our approach consistently achieves comparable even better performance than supervised NAS methods on different search spaces and datasets. Taking S4 as an example, our approach discovers the model with the lowest error rate of 16.54% on CIFAR-100, which outperforms other methods with a clear margin.

## 4.7 GENERALIZATION ABILITY

We verify the generalization ability of the proposed method on downstream tasks. Specifically, we transfer different NAS models searched and pre-trained on ImageNet to the detection task for fine-tuning. RetinaNet (Lin et al., 2017) and MS COCO dataset (Lin et al., 2014) are chosen as the

Table 9: Object detection results of DARTS search space on MS COCO.

| Method | Params (M) | FLOPs (M) | $AP$ | $AP_{50}$ | $AP_{75}$ |
|---|---|---|---|---|---|
| Random search | 4.7 | 519 | 31.7 | 50.4 | 33.4 |
| DARTS | 4.7 | 531 | 31.5 | 50.3 | 33.1 |
| P-DARTS | 4.9 | 544 | 32.9 | 51.8 | 34.8 |
| PC-DARTS | 5.3 | 582 | 32.9 | 51.8 | 34.8 |
| UnNAS (rotation task) | 5.1 | 552 | 32.8 | 51.5 | 34.7 |
| RLNAS | 5.2 | 561 | 32.9 | 51.6 | 34.8 |
| Ours | 4.7 | 533 | **33.0** | 51.8 | **35.1** |

backbone and validation dataset for the detection task. To ensure a fair comparison, we follow the same training setting as RLNAS for both pre-training and fine-tuning. The only difference is that we replace the backbone of RetinaNet with the model searched by our approach. Table 9 summarizes the comparative results, showing that our searched model on the DARTS search space achieves higher AP on the COCO dataset.

## 4.8 SENSITIVITY ANALYSIS OF MASK RATIO AND PATCH SIZE

In MAE, mask ratio and patch size are two important parameters, which will greatly affect the modeling. Mask ratio refers to the proportion of pixels in an image that are randomly masked or hidden during the training process. This masking process helps the model learn robust representations by forcing it to reconstruct the original image from incomplete or corrupted inputs. Patch size, on the other hand, refers to the size of the masked patches in the image. These patches are randomly selected and masked during training, and the model is trained to reconstruct the original image from the remaining unmasked pixels. The patch size determines the spatial extent of the masked regions in the image. We are currently evaluating the sensitivity of our method to these two parameters. Table 10, shows that these two parameters have a minimal impact on the final search results.

Table 10: Searching performance on CIFAR-10 in S0 w.r.t the mask ratio and patch size.

| Mask Ratio | Top-1 Acc (%) | | Patch Size | Top-1 Acc (%) |
|---|---|---|---|---|
| 0.1 | 2.84±0.22 | | 2 | 2.75±0.24 |
| 0.3 | 2.77±0.14 | | 4 | 2.63±0.11 |
| 0.5 | 2.65±0.08 | | 8 | 2.71±0.12 |
| 0.7 | 2.80±0.31 | | 16 | 2.80±0.26 |

## 5 CONCLUSION

By leveraging labeled data, NAS algorithms can extract meaningful patterns, leading to state-of-the-art architectures. However, obtaining labeled data is costly and time-consuming, making unsupervised NAS methods attractive. We propose a NAS framework based on Masked Autoencoders that eliminates the need for labeled data during the search process. Our approach replaces the supervised learning objective with a reconstruction loss, enabling the discovery of network models with stronger representation and improved generalization. Experimental results on seven search spaces and three datasets demonstrate the effectiveness of our method, achieving comparable accuracy under the same complexity constraint.

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
