# OpenReview forum: "Masked Autoencoders Are Robust Neural Architecture Search Learners"
_ICLR.cc/2024/Conference — ICLR 2024 Conference Withdrawn Submission_

### Official Review · Reviewer_eDJx · 2023-10-24

**Soundness:** 3 good
**Presentation:** 3 good
**Contribution:** 2 fair
**Rating:** 5
**Confidence:** 4

**Summary:**

This paper devised a new unsupervised neural architecture search algorithm, where a masked auto-encoder is used for reconstructing the input image. They adapted DARTS algorithm and instead of training the supernet to maximize the validation accuracy, the supernet is used as a backbone in the encoder. The decoder uses the  latent features of varying scales output by the encoder outputs and reconstructs the image.
 They authors observed that their method also suffers from dominant skip connecting if the masking ratio is less than 0.5. It is however alleviated at higher values.

**Strengths:**

This is a new unsupervised NAS algorithm and it performs as well as the other supervised NAS algorithms.

**Weaknesses:**

1. Can you demonstrate that the proxy task chosen is effective by ranking the architectures in increasing order of reconstruction error with the true ranking of the architectures trained from scratch on the validation accuracy of the classification task?
2. In order to show that your method is generalizable, can you also include the experiment performed by Unnas where the architecture was searched on a different source dataset and evaluated on a different target dataset?

**Questions:**

1. Given that there already exist two unsupervised methods and your technique performs very similar to those two, what is the advantage of using your method when compared to theirs? Please elucidate it

---

### Official Review · Reviewer_qAFC · 2023-10-31

**Soundness:** 2 fair
**Presentation:** 2 fair
**Contribution:** 2 fair
**Rating:** 3
**Confidence:** 5

**Summary:**

This paper leverages masked autoencoder to perform unsupervised differentiable neural architecture search on CNN. The author design a multi-scale decoder to solve the performance collapse issue in differentiable neural architecture search methods. The experiments are conducted on nas201, imagenet, and coco,

**Strengths:**

This paper investigates the combination of mae with neural architecture search, which is interesting and worthy to explore.

**Weaknesses:**

1. The experimental results of the proposed method are, in fact, very bad. In both Table 2 & Table 3, the darts-minus performs better or on par than mae-nas. In Table 9, there is no comparison to darts-minus, but rlnas is also very competitive, only 0.1 AP worse than mae-nas with less than 5% computation budget. These results indicate mae-nas is not effective.

2. It does not make sense that mae-nas can be applied to CNN, since previous works [1] have shown that masked unsupervised learning is not working on CNN unless with complicated designs. The author need to explain it.

[1] ConvMAE: Masked Convolution Meets Masked Autoencoders

3. No correlation analysis on mae score versus accuracy in this paper. This is an necessary part in the NAS paper.

**Questions:**

See weakness.

It is very **surprising** that in Table 5, the (best) results for MAE-NAS are **exactly the same as** the best results of DARTS-, over 4 runs. It is reasonable to doubt that the author simply copy-paste the results of DARTs on MAE-NAS.

From my point of view, the experimental results are too weak to support the idea that MAE is a valid approach for neural architecture search.

---

### Official Review · Reviewer_CagH · 2023-10-31

**Soundness:** 2 fair
**Presentation:** 2 fair
**Contribution:** 2 fair
**Rating:** 3
**Confidence:** 4

**Summary:**

This paper introduces masked autoencoders (MAE) into DARTS methods to optimize the unsupervised reconstruction objective instead of the supervised classification objective. The motivation of the paper is that the unsupervised MAE eliminates the need for a large volume of labeled data. The authors also propose a multi-scale decoder to enhance the MAE reconstruction. Experimental results on CIFARs and ImageNet with DARTS variants are provided.

**Strengths:**

The proposed method of unsupervised MAE + DARTS is simple and easy to understand.

**Weaknesses:**

1. Overall, the improvements across almost all tables are marginal, with some of them even underperforming the existing methods. Moreover, the proposed method yields a larger std than DARTS on average in Table 5. As is known, UnNas should be the baseline of the unsupervised NAS. But, as shown in Table 4, it is hard to clearly say the proposed method is better than UnNas. Thus, the experimental results make the contribution of the paper limited.

2. The authors claim the multi-scale decoder stabilizes the training process and prevents collapse in DARTS optimization. However, the paper lacks of detailed discussion and ablation study to evidence the claim.

3. Given that the combination of MAE + DARTS is quite straightforward and the experimental improvements are trivial, the novelty of this paper may be marginal.

**Questions:**

Please see the weaknesses above.

**Minor questions**
- Why use the question mark in the title?

---

### Official Review · Reviewer_UdAV · 2023-11-02

**Soundness:** 2 fair
**Presentation:** 3 good
**Contribution:** 2 fair
**Rating:** 5
**Confidence:** 3

**Summary:**

This paper introduces a novel Neural Architecture Search (NAS) method, MAE-NAS, which replaces the supervised learning objective with an image reconstruction task, effectively eliminating the need for labeled data in the search process. The authors address the challenge of performance collapse in the Differentiable Architecture Search (DARTS) methodology by implementing a multi-scale decoder. Extensive experiments across various search spaces and datasets validate the effectiveness and robustness of MAE-NAS, confirming its ability to discover efficient, well-generalizing network architectures without labeled data.

**Strengths:**

Through the authors' results, I find the important or novel discovery to be a good network for image reconstruction can also be a good network for classification. This knowledge is relatively novel to me. Considering MAE's results, the linear features learned from MAE's unsupervised extended schedule are not outstanding compared to MOCO or other unsupervised tasks. However, after fully fine-tuning, the model's performance shows MAE can scale better. This paper also confirms this through a NAS perspective. The results are not that strong but the author examines their searched networks in various benchmarks and shows effectiveness. Also, the paper is easy to read.

**Weaknesses:**

While the experimental results in the paper primarily showcase the superiority of the searched model across various benchmarks and scenarios, what intrigues me, however, is a more in-depth exploration of **why** and **how** Masked Autoencoders (MAE) aid DARTS. More specifically, I'm interested in understanding the circumstances under which MAE works well and where it might not.
1. How does MAE-NAS perform on larger-scale data, such as ImageNet?
2. What is the quality of the reconstruction? Is this related to the searched model's performance?
3. Are there any conducted ablations about different levels of reconstruction losses, and what is their impact on the models searched?
What is the most influential factor in finding a better architecture using MAE-NAS?

A good showcase example of the above questions is "Are Labels Necessary for Neural Architecture Search?" (see Fig.2~5),  and I believe more comparisons about this paper's method should be conducted to show differences and superiority. Otherwise, from Table 4, the conclusion can be too simple and lack insights.

Also, while the current test was apparently done on small-scale data, one of the key advantages of unsupervised methods is the ability to scale up learning on a much larger scale without human annotations. Therefore, understanding how the scale-up problem is addressed within MAE-NAS should be considered.


[1] Liu, Chenxi, et al. "Are labels necessary for neural architecture search?." Computer Vision–ECCV 2020: 16th European Conference, Glasgow, UK, August 23–28, 2020, Proceedings, Part IV 16. Springer International Publishing, 2020.

**Questions:**

See above weakness.